# Working with Data in Adult English Classrooms: Lessons Learned about Communicative Justice during the COVID-19 Pandemic

**DOI:** 10.3390/ijerph20010696

**Published:** 2022-12-30

**Authors:** Margaret A. Handley, Maricel G. Santos, María José Bastías

**Affiliations:** 1Department of Epidemiology and Biostatistics, University of California San Francisco, San Francisco, CA 94158, USA; 2Department of English, San Francisco State University, San Francisco, CA 94132, USA

**Keywords:** COVID-19, English as a Second Language (ESL), communicative justice, data visualization, data literacy, learner leadership, community-based participatory research, health communication

## Abstract

Throughout COVID-19, health officials have relied on data visualizations to communicate urgent messages about the spread of the virus and preventative measures. Relatively few efforts have employed participatory engagement with communities who have experienced a disproportionate burden of COVID-19 illness to shape these communications. Sociologist W.E.B. Du Bois viewed data visualization as an approach to *changing the way people think about themselves*. This paper describes a community-engaged approach to data literacy skill-building with bilingual Latina learners in an adult English program in Northern California, Bay Area. The curriculum combines data visualization activities with language instruction and preventive health themes. Early work on COVID-19 in 2020–21 emphasized improving health knowledge and message interpretation but later shifted to a critical data literacy perspective, focusing on myth-busting, improving risk messaging in their own social networks, and supporting learners to see the power of their own experiences in data story-telling processes. This pedagogical approach, guided by Charles Brigg’s idea of *communicative justice* priorities, locates adult learners’ data visualization work as part of a broader effort to be included in the perspectives that shape knowledge production in today’s healthcare system. This approach can be used to examine disparities in information access in linguistically minoritized communities and guide future education interventions.

## 1. Introduction

COVID-19 has dramatically altered the way we create, disseminate, and make sense of risk data and risk communication. Unfortunately, the majority of COVID-19 prevention-focused images in the form of data visualizations do not take into account language and literacy barriers (including low levels of health literacy, data literacy and English proficiency) that may limit interpretation in key population groups. Equally disturbing is the absence of public health images generated through engagement with those experiencing a disproportionate burden of COVID-19 illness, such as Black and Latinx populations in the United States [1,2,3]. In this article, we look at the opportunities and challenges the COVID-19 pandemic has created for community engagement as a data science strategy by focusing on the adult English language classroom.

The U.S. adult English language learner (ELL) population comprises nearly half (43%) of the approximately 1.1 million adult learners served by the U.S. publicly funded adult education system [4]; this number does not include thousands of learners enrolled in community-based programs supported through other mechanisms, e.g., private foundations. The U.S. ELL population includes immigrants, refugees, parents, the elderly, “working poor” adults (those under-employed or seeking work), and youth who have aged out of the K-12 system. An increasing number of learners in the system include adults who have not learned to read or write in any language and/or have experienced little or interrupted formal schooling [5,6].

A classroom of adult English language learners is not often the image that comes to mind when people think of data scientists, but we argue that these classrooms are a neglected resource in efforts to democratize data science and data and health-related literacies. The healthcare experiences of linguistically minoritized learners should be part of the knowledge base that informs what we know about the interconnectedness of language, power, and health, as well as the way these lived experiences shape engagement with COVID-19 risk messaging. Adult learners have their own interpretations of health information circulating around us—the data, images, media campaigns, and sound bites—yet they could rarely share their data interpretations or their needs for data with other learners, local communities, or public health researchers. To date, the adult literacy classroom has received relatively little attention as a legitimate makerspace where linguistically diverse communities can work with data tools and data interpretation/production practices and thus be recognized as valued contributors to the world of community-engaged data science.

### Conceptualizing Data Literacy

As educators, public health practitioners, care providers and communities grapple with how to boost the public’s capacity to navigate diverse sources of data related to health, including misinformation, the arena of health literacy has expanded in the last decade to include a range of spin-off literacies in the data realm: digital literacy, digital health literacy, data literacy, data visualization literacy; and, more recently, consumer literacy, media literacy, and creative data literacy. Some scholars approach the *conception* of data literacy as a competency to be achieved at the individual level, such as with digital literacy or the numeracy component of health literacy, with an emphasis on comprehension [7]. Other scholars focus on visualizations of data, emphasizing the *attributes of images* that may increase patients’ engagement with health communication tools such as with patient-decision aids or medication adherence apps [8,9,10]. Concepts such as *risk literacy* focus on people’s understanding of risk data in particular (e.g., probability statistics) and the benefits and limitations of visual aids as health risk communication tools [11].

Curriculum standards related to data literacy have been pivotal in directing attention to the role of K-12 schools and higher education in skill-building, although competencies related to data interpretation and visualization are described in different ways. For example, in California’s Common Core Standards the ability to work with information in “diverse media and formats (e.g., visually, quantitatively, orally)” is part of oral and written communication skills [12]. The International Standards for Technology Education (ISTE) Standards for Students describes the ability to “sift through data” and find patterns in numbers, texts, and pictures as key “Computational Thinker” standards [13]. The Northstar Digital Literacy framework, specifically developed for the adult basic education system, lists some data visualization skills under Standards for Essential Software Skills (e.g., “Sort (least to greatest, alphabetically, etc.) and filter data”) and Standards for Using Technology in Daily Life (e.g., “Identify types and formats of information found online (articles, databases, images, videos, etc.)” [14]. These frameworks share an emphasis on collaborative learning and the transfer of skills to real-world problem solving [15].

A more collective conceptualization, one that may relate to community empowerment, shifts attention to how communities can be involved in the creation and interpretation of data, a participatory approach that has emerged from environmental justice initiatives to explain and address disparities and pollution risks [16,17]. In this work, there is not only an emphasis on individuals’ understanding of their own exposure risks but also a focus on elements of the data visualization process that enable community members to work with local risk data and advocate for changes to reduce those risks. Such concepts build on the work of anthropologist Charles Briggs, who views the dismantling of inequities in power and health communication pathways—along with the localization of interpretive authority in communities—as central to the advancement of “communicative justice” in public health and medical care [18].

A related concept focuses specifically on the act of data visualization as itself an important driver of knowledge production and agency in communities [19]. For example, W.E.B. Du Bois, a well-recognized early pioneer in sociology, particularly African American Sociology, is under-credited for advancing data visualization as an approach to *changing the way people think about themselves* through *the process* of creating visualizations [20]. This process can be considered both a form of creative data literacy skill building and participatory engagement, with its focus on participatory data collection.

The promise of W.E.B. Du Bois’s vision is realizable when the tools needed to “render information in a visual format” are also accessible to community members. This argument is powerfully made in the participatory data visualization work by Catherine D’Ignazio and colleagues [19,21]. D’Ignazio cites “a growing gap between those who can work effectively with data and those who cannot,” further arguing that when only “state and corporate actors…possess the resources to collect, store and analyze data, individuals (e.g., citizens, community members, professionals) are more likely to be the subjects of data than to use data for civic purposes” [21]. This gap in access is particularly acute for patient populations commonly described as “limited English proficient” who have disproportionately suffered during the COVID-19 pandemic [22]. Exacerbating these health disparities are inequities in risk communication where linguistically minoritized communities are less likely to access opportunities to learn how to work with data, particularly in a socio-political climate where English is viewed as the default language of academic authority and data science expertise. Our pilot work ultimately aims to fulfill W.E.B. Du Bois’s vision of data visualization as a powerful act of discovery and D’Ignazio and Bhargava’s work on creative data literacy as a means to reduce the gap between civic society and those with data knowledge.

In this article, we share our pedagogical approach and lessons learned with Latina women in community-based ELL classrooms before and during the pandemic, including implications for future equity-driven interventions in data science. The classes taught by María José Bastías (third author) incorporate a participatory approach to data literacy and data visualization that emphasizes shared inquiry into data sources and creative interpretive elements (e.g., using role play to explore different interpretations of data) and reflect a social constructivist approach to how knowledge is created, which we have used in our classroom-bases work previously [23,24]. This emphasis on the learning experience stands in contrast to the emphasis on what is a “good picture” or how to “gather numeric data” from a data visual, which is more widely encountered in discussions of data visualization and data literacy. A solidarity around the act of interpretation emerges in the classes (e.g., “how does this bar graph help our community know where to go for the vaccine?”) as the women use their linguistic resources to test out interpretations, bridge gaps in understanding, and assess relevance to their everyday lives.

## 2. Materials and Methods

### 2.1. Theoretical Approach

Social constructivism is a sociological theory of knowledge that focuses on how individuals come to create and apply knowledge in socially mediated contexts. The fundamental premise of this theory is that knowledge is a human construction and that learners are active participants in the learning process [25,26,27,28].

### 2.2. Pedagogical Approach: Learning, Language Learning, and Learning to Work with Data

Learning to work with data, like any learning process, is a deeply contextualized activity. While learning to use and analyze data obviously involves the acquisition of cognitive skills and technical know-how, learning to work with data is also a function of developing shared social practices [29,30]. From a social constructivist perspective, data literacy is best viewed as an interactional competence [31,32]; data literacy tools and processes become accessible to adult learners through their re-occurring interactions with those tools and processes with others. From this view, learner language is viewed as a critical driver of growth in interactional competence. D’Ignazio and Bhargava have also recognized the role of language in participatory engagement processes in their calls to support community members in being able to “speak data” [19]. We posit that, in the context of data literacy, learners need opportunities to learn to “speak data” as part of three interdependent, co-occurring processes:Learning language (e.g., new vocabulary such as “percentages”),Learning new data skills *through* language (e.g., talking with others as they puzzle through data points), andThinking and talking *about* the language they are using (e.g., “40% women feel stress. Should we say ‘a lot of women’, or ‘some women’? And the women—who are they? What age? Where do they live? Are they like us?”).

Supporting learning in all three areas is essential to growth in learners’ interactional competence as data users.

For linguistically minoritized communities for whom English (the language of greatest currency in data science) is not yet a dominant language, we argue that “speaking data” must also be understood as a phenomenon of *translanguaging*, referring to the way emerging bilinguals use the languages they already know (everyday language, first languages, science language, math language, imperfect English, etc.) to mediate cognitively complex tasks, such as learning to interpret a pie chart or sort data. Canagarajah has conceptualized translanguaging as “the ability of multilingual speakers to shuttle between languages, treating the diverse languages that form their repertoire as an integrated system” [33] (p. 401). This linguistic shuttling is an overlooked interactional resource in learners’ efforts to “speak data”, as learners use languages already accessible to them to problem-solve, test out interpretations of specific data or visual elements, and exchange emotional responses to data. Hellermann further argues that “the greater the degree to which meaning is shared within communities, the greater the potential for participation for all those members who share in that ownership” [34] (p. 12). From a translanguaging perspective, there is no inherent hierarchy in the languages used; all are considered critical meaning-making resources.

This combined perspective on language, language learning, and data literacy learning is crucial to identifying the kinds of learning spaces that can support this kind of growth. Arguably the community-based classroom is one of the few contexts where linguistically minoritized adults are encouraged to experiment with and practice “speaking data”, and mistakes are considered opportunities for learning.

### 2.3. Preliminary Classroom Activities

Prior to the pandemic shutdown, we (MS and MH) had been collaborating with Bay Area adult literacy teachers, including María José Bastías, to develop classroom activities that combined data visuals and English language learning. The activities emphasized supporting learners to understand and use language needed to interpret risk communication, such as the language of frequency and probability (e.g., *likely, less probably, high/low risk*), and the language of proportion (e.g., *1 out of 7 women*). For several of the classes, we analyzed recorded transcripts to create word clouds (Figure 1 and Figure 2) and to provide examples of translanguaging collaborative work by students.

Building on our prior work with English language learners and clinic-based populations at elevated risk for type 2 diabetes [23,24,35], we developed activities in which learners explored heat maps depicting the sale of sugary drinks and rates of Type 2 diabetes, by zip code, for their own neighborhoods and other surrounding areas. We presented heat map data in a simplified, easy-to-read format so that learners could role-play language for interpreting the heat map data such as “What does orange mean on the map?” and “Orange means more people have diabetes in this neighborhood compared to people in other neighborhoods”. Perhaps more importantly, we observed the value of supporting learners in *seeing themselves in the data*, to locate their own neighborhoods, and explore whether the data affirmed or tested their own perceptions.

### 2.4. Data Literacy Pedagogy and COVID-19

When the pandemic shut down classrooms in Spring 2020, the idea of learners “seeing themselves in the data” took on new meaning. Beginning in March 2020 and extending for the next 9 months, María José shifted to teaching online modules lasting 3–5 weeks focused on learners’ desire to understand COVID-19 trends and develop counter-narratives to the widespread COVID-19 misinformation they were encountering in their daily lives, such as that there were no requirements of employers to provide masks or to use appropriate ventilation in close workspaces. Based on course evaluations and María José’s own observations, learners were motivated to learn English, but they also sought out the classroom community for refuge, support, and health information during the pandemic. With the release of new COVID-19 trends in the news and changes in public safety guidelines, lesson themes evolved as learners shared their fears, discoveries, and interests in learning more. These classroom activities around COVID-19 themes and different types of data prompted learner exploration of other “hot topics” which, over the course of 2020, included interpretation of public health messages, labor rights, advocacy and social justice, questioning sources of data, risk prevention, public health regulations in the workplace, navigating power dynamics at work and home around social distancing during COVID-19, and the role of technology in daily life.

For the free data literacy-focused modules on COVID-19, we set up online interactive English as a Second Language (ESL) classes through a non-profit Latinx organization located in the Northern California Bay Area, in collaboration with Maricel Santos (adult literacy researcher, San Francisco State University) and Margaret Handley (public health researcher, University of California San Francisco). The curricular scope of pre-pandemic, as well as pandemic-specific modules, included three related components (see Section 2.5, Section 2.6 and Section 2.7).

### 2.5. Language Learning and Preventive Health Themes

In the bi-weekly sessions, each lasting 4–6 weeks with 20 h of instruction, learners gather over Zoom and are guided through data story-telling processes [23]; they are invited to explore interpretations and visualizations based on information they collect amongst themselves and across their social networks which they share asynchronously on Facebook. Class activities and curriculum were geared towards developing confidence and competence as multilingual speakers of Spanish, Mam (an indigenous, Central American language), and English. During class, women analyzed, learned, and employed speech acts, utterances that serves a particular communicative function in discussions about data and data interpretation, such as speculating, providing arguments based on reliable information and their experience, and reaching agreements. When learners linked data to real-world actions (e.g., using data about safety protocols to ask their boss for personal protective equipment, such as masks), they were able to practice speech acts used to negotiate, make requests, accept/refuse offers, and suggest solutions. Learners also carried out fieldwork-type activities, such as daily language observations of the ways English, Spanish, and Mam are used in different social and institutional contexts. These inquiries help learners name the inequities associated with the dominance of English over other languages in public communication, and as a counterweight, learners are encouraged to see all known languages as valid resources for sense-making.

### 2.6. Data Literacy Skill Building in and beyond the Classroom

Building on participatory engagement methods, the classroom activities were designed around asking questions and exploring data that could be used to explore answers, collecting data in informal networks, and then “re-storying” the information collected, through classroom discussions [23]. As highlighted in Table 1, the learners were introduced to different sources of data (e.g., structural determinants data, prevalence data, risk assessment data). The learners gained hands-on experience with data tools, most frequently through Google tools, but also through low-tech materials (e.g., markers, paper plates for making pie charts). Women work as a group to visualize their findings, using Google forms, hand-drawn social network maps converted to presentations, and free online software. Then, women share their class products (e.g., slide presentations, digital collages, or short videos recorded on smartphones).

### 2.7. Understanding Social Networks and How to Engage Them for Data Collection and Health Messaging

One area of data literacy skill building involved activities aimed at improving the women’s analysis of their own informal social networks and how they do and do not reflect other channels of communication, especially for health-related information. Activities included developing survey questions and formatting them to share over WhatsApp, collecting data across all participants to obtain a classroom network response (a form of respondent-driven sampling), and summarizing the results as a group activity. Women also developed a range of visualizations to characterize their social networks (Table 1; Figure 3 and Figure 4), adapting established methods to engage with populations who may have limited engagement with traditional health sector services or who are otherwise not well understood. Learners’ engagement with social network mapping was also leveraged to the ‘pushing’ out messages developed in the classroom, where learners shared their data visuals with family and friends.

## 3. Results

In this section, we report results from the November–December 2020 and Fall 2021 instructional cycles, which include a focus on vaccination.

### 3.1. The Classroom Participants

The classroom demographics reflected the community affiliated with the non-profit Latinx organization; the women were originally from Central America, predominantly from Mexico, El Salvador, and Guatemala. Most worked in service industries as child, patient, and senior caregivers, janitors, restaurant workers, housekeepers, homemakers, among others. The majority of the women (75%) were between 20 and 49 years old, with another 25% between 50 and 70 years old.

Classes were advertised through Facebook, word-of-mouth, individual phone calls to invite and follow up with women, and brief invitations during the organization’s general community meetings. The class registration was performed through Google forms and the Zoom links were sent to participants emails and WhatsApp groups. On average, participants attended between 16 and 18 h of instruction over a 2 to 4-month period. Notably, each module enrolled women who, prior to the pandemic, had never enrolled in the organization’s English classes. In some cases, these new enrollments were a function of increased accessibility to classes because of the Zoom platform; in other cases, women started attending classes after pandemic-related job losses. All women were new attendees to the ESL classes offered by this organization. Although attendance was not mandatory, learners were eager to join the online classes and learn how to use technology to communicate with others, either creating or receiving messages. For example, they let the instructor know when they could not be in class, they joined while running errands, or arranged their work breaks and shifts to be able to participate.

Three hundred women registered, representing 11 cities in the San Francisco Bay Area. Eighty (26.6%) attended at least one out of seven classes; 70 attended at least two classes. Close to two-thirds of participants (n = 50/80) created at least one original visual (e.g., word clouds, video with composite images, bar graph, bubble or ‘heat’ map).

### 3.2. Student-Led Social Network Surveys and Data Visualizations

Participants reported sharing classroom work with family and friends in home countries. Figure 1, Figure 2, Figure 5 and Figure 6 depict examples of data visuals that integrate learner-generated data. In Figure 5, a survey administered by students to members of their social network about how COVID-19 has affected them led to a bar graph of the main results, and a discussion of the concepts as well as the results. For example, the bar graph represents all responses and answers, which are not mutually exclusive, so a total can be greater than 100%. Students then explored how to take responses from the bar graph and transform it into a word cloud to compare the different modes of presentation (data not shown).

Figure 6 represents a pie chart from the same student-led survey and conveys the results about COVID-19 test information, which led to a discussion about the relationship between words that convey size (many, majority, some, few) and different ways these might be seen in numeric data as percentages or frequencies and what level of detail is enough to make a decision about taking a specific type of test.

The following exchange represents the way the discussion of data interpretation is providing an opportunity for multilingual and multimodal interpretation of information.


*Original (Bilingual, Spanish & English)*


Teacher:¿Qué piensa la gente? So COVID-19 tests are free? Son gratis. What do people think?

Learner 1:Aquí si son gratis.

Teacher:La mayoría opinó que si son gratis. Entonces cómo decimos “*la mayoría*” en inglés?

Learner 1:more people?

Teacher:… most people

Learner 1:*oh most people… most people*. ¿Cómo se escribe *most people*?

Teacher:Give me one sec, I’m typing on the chat. Ok. Most people think that COVID tests are free. Si se dan cuenta en español decimos *“la mayoría de la gente piensa”* son 6 palabras, en inglés ¿cuantas vamos a usar?

Learner 1:*most… people… think*… one, two, three.. Three?

Learner 2:most people think, son tres, three words.

Teacher:ahora, de este grupo, ¿qué porcentaje piensa *tienen que pagar por un COVID test?*

Learner 3:five por cent?

Teacher:yes, five percent. So, 5% de las mujeres que tomaron esta encuesta piensan que hay que pagar por un COVID test. What does it mean? Qué significa esto?

Learner 3:por no estar informada

Learner 1:no están bien informadas

Learner 4:que no están bien informadas

Teacher:claro, first, *they don’t have **enough** information* (types sentence and shares on chat). Miren, pongan atención en la oración que está en el chat, miren la palabra *enough* (models pronunciation and learners repeat). ¿Qué más significa que alguien crea que tiene que pagar por un COVID test? Are they going to take it?

Learner 2:probably not


*Translated to English*


Teacher:¿What do people think? So COVID-19 tests are free? They are free. What do people think?

Learner 1:Here they are free

Teacher:Most people answered that they are free. So how do we say “la mayoría” in English?

Learner 1:more people?

Teacher:… most people

Learner 1:oh most people… most people. How do you spell most people?

Teacher:Give me one sec, I’m typing on the chat. Ok. Most people think that COVID tests are free. If you notice in Spanish we say “la mayoría de la gente piensa”, that’s… 6 words. In English, how many words are we going to use?

Learner 1:most… people… think… one, two, three.. Three?

Learner 2:most people think, that’s three… three words.

Teacher:Now, from this group, what percentage think that you have to pay for a COVID test?

Learner 3:five percent?

Teacher:yes, five percent. So, 5% of women that answered this survey think that they have to pay for a COVID test. What does it mean? Que significa esto?

Learner 3:for not being informed

Learner 1:they are not well-informed

Leaner 4:that they are not well-informed

Teacher:Right, first, they don’t have enough information (types sentence and shares in chat). Look, pay attention to the sentence in chat, look at the word enough (models pronunciation and learners repeat). What else could mean that a person believes they have to pay for a test? Are they going to take it?

Learner 2:probably not

Using COVID-19, the lens of the classroom initially incorporates individual determinants, such as views about masks and social distancing, and then incorporates structural challenges learners may face in the workplace, such as negotiations about prevention behaviors in their jobs (e.g., speaking up, mask wearing, social distancing). Teachers guide women to ask and answer their own questions, and importantly, to interrogate who owns the right to create, present, and interpret data and call out its truthfulness. Additionally, women collaboratively solved problems of communication by giving each other advice, suggestions, and information about available resources.

Fourteen of the fifteen women enrolled in the November-December classes in 2020 developed a short survey and completed the respondent-driven sampling activity to determine behavioral intentions among family and close friends about COVID-19 prevention measures over the holidays. The rapid survey focused on the following questions: “How many people will you see over Christmas?” (count data); “How will you protect yourself?” (categorical data); and “what are you most afraid of?” (open-ended data). The women collectively received 76 responses in Spanish (5 per woman) in 36 h using Facebook and WhatsApp and survey responses were summarized in class with bar charts to estimate the risk behaviors anticipated over the holidays, and to generate a group discussion of strategies to reduce the risks. Survey respondents were primarily Latino (97%), women (93%), with the majority ages 30–39 (39%) followed by 31% 40–49, 17% 50 and older and 13% under age 30. Most lived in the San Francisco Bay Area (82%) with 13% living in the Central Valley of California and the rest out of California.

Learner-identified concerns included getting infected by someone who is asymptomatic (50%); 30% were worried others would not take adequate precautions and 7% were worried they might infect others without knowing. Risk prevention strategies encompassed maintaining social distances inside (51%), mask wearing indoors (41%), not visiting in the first place (41%), improved ventilation (37%), not sharing utensils (37%), staying outside (17%), and social isolating the week before (17%). These results were discussed in class and also shared with local health department COVID-19 response teams.

Overall, and across classes, learners disseminated their visuals to co-workers, and family and friends in home countries using WhatsApp and Facebook. Four of the bilingual visuals posted to a community Facebook page each received between 250–500 views. Visual themes on COVID-19 included: avoiding fraud schemes, supporting children; worker rights; and safety for domestic and essential workers. Women also shared their newly created images, videos and public health safety stories via their WhatsApp networks. UCSF collaborators also shared the original health messaging videos and images with local and state public health departments to aid in shaping local responses to COVID-19 prevention barriers (such as mask wearing by co-workers and employer workplace safety).

### 3.3. Assessment of Classroom Examples

As discussed earlier, central to the learners’ data skill-building is the opportunity to “speak data” [19]. Table 2 provides examples of learners’ data interpretation and data visualization activities, including examples of their own contributions to ‘speaking data’ in the classroom for the Fall 2020-Fall 2021 instructional cycle. These examples provide promising evidence that adult learners in these community-based settings engage with data interpretation and visualization choices that parallel what technical experts do and say. The examples in Table 2 also suggest that, for the learners in these community-based settings, learning to “speak data”—literally, finding the “right” words to describe what they see *in* data and data visuals—is not only a critical component to data–skill building; it is a vital act of inquiry in and of itself.

We also emphasize the inquiry mindset that seemed to arise from these scaffolded learning opportunities and data interpretation processes. Based on our classroom observation and teacher notes, we saw positive changes not only at the individual level (e.g., increased motivation, confidence with data sources), but also at the classroom community level as the learners identified areas of collective action that arise from their data analytic work, such as:Increased learner confidence with scientific and everyday ways of talking about data through collaborative translation and paraphrasing of data interpretations and preparing them on their own.Increased motivation and familiarity with tools and processes for developing and implementing a data collection tool among social networks and to visualize the results.Increased awareness of their own preventive health thinking skills, specifically to link their data visuals to action steps needed in their communities.

Thus far, we have relied on learner self-report and our own teaching observations to track growth in these areas, but future iterations should formalize a battery of classroom-based assessment tools that capture the impact of data literacy skill-building at the learner and collective (classroom/community) levels.

Towards the end of the first year of the pandemic (Fall 2021), four of the women learners co-authored a bilingual commentary for the organization’s newsletter, with guidance from María José, and reflected on their data work. Notably, by the beginning of the 2021–2022 school year, the four women who worked with María José on this bilingual commentary had enrolled in non-credit child development/English courses at a local community college, in part inspired by their increased confidence working with new forms of information and research tools. Below is what appeared in the bilingual commentary:

Original Spanish:

Aprendemos acerca de nuestros derechos laborales, nosotros practicamos como pedir ayuda para mejorar nuestra salud mental y física…. Interpretamos datos como gráficos de barras, gráficos circulares, nube de documentos de word, estadísticas, evaluar fuentes de información; y evitar fraudes y estafas relacionadas con COVID-19. En nuestra clase no hay respuestas incorrectas lo cual nos da la confianza de NO tener miedo cuando hablamos frente a otros… nos sentimos más confiadas y empoderadas.

English translation:

We learn about our labor rights; we practice asking for help to improve our physical and mental health for us and our families. We interpret data such as bar graphs, pie charts, word clouds, statistics; we evaluate sources of information and learn to avoid fraud and scams related to COVID-19. In our class there is never a wrong answer, which provides us with the confidence to not be afraid when talking in front of others… we feel more confident and empowered.

The women’s words call attention to the learning conditions that appears to be central to growth in their interactional competence with data: access to tools (e.g., “bar graphs, pie charts”), linking data practices with real-world action (e.g., “avoid fraud and scams related to COVID-19”), and attention to the emotional labor in data work (e.g., “the confidence to not be afraid talking in front others”).

## 4. Discussion: Lessons Learned

Our work with Latina women in community-based, English language learning classrooms, where the English classrooms combine instruction with preventive health themes (e.g., stress reduction, COVID-19 prevention) and data literacy skill-building, emphasizes data interpretation as a social and creative activity [36].To our knowledge, this is the first study exploring the unique constellation of making meaning from a range of emerging and often conflicting data during a pandemic that also applies translanguaging and collective participatory data analysis. The collaborative translanguaging—the puzzling-through the meaning in a data visual (e.g., a COVID-19 prevalence heat map) across multiple languages (e.g., Spanish, English, Mam)—can deepen conceptual understanding of risk, an opportunity rarely afforded to linguistically minoritized populations in official English-dominant settings. Lessons build on questions posed by learners, who in turn poll their peers for responses (e.g., “What do you do if your co-worker isn’t wearing a mask?”), as well as essential questions about data literacy: What is data? What data do you trust? From a translanguaging perspective [37], translations across languages (e.g., “what is ‘flatten the curve’ in Spanish?”) are in service of knowledge-building and inclusive conversations. These findings illustrate the kind of community-owned inquiry that drives this data-intensive work, along with the various data types, data visualizations, and dissemination outlets that the learners found accessible.

This kind of curricular innovation requires attention to learner needs related to connectivity and digital readiness. The pivot to remote instruction enabled greater numbers of Latina women to participate than are able to regularly attend in-person class sessions. At the same time, poor broadband connectivity hampered regular participation of Spanish-Mam speaking women, older women, and those with limited print skills who seemed more reluctant to create visuals on their own. To improve sustainability, language and literacy programs may benefit from establishing core data literacy competencies [14]; they may also benefit from addressing the need for digital skill-building, including trust in data technologies, to ensure such materials reach a diverse audience and support participation across modalities.

Our classroom work during the pandemic suggests that data literacy competence is a multilingual, multimodal achievement. Learners work to make sense of “ensembles” of health information coming at them in a variety of modes: print, speech, images, graphics, and sound [38]. Our work with this community of women learners provides clarifying insight into the ways people move across modalities (print, audio, visual, spoken), tools (markers and paper, smartphones, tablets, apps) and languages (English, Spanish, Mam) in their sense-making processes. We have found that the women’s most incisive interpretations of COVID-19 data visuals are not fixed to any one language; rather, it appears that the movement across languages and modalities strengthens their risk comprehension, cultivates trust in the message, and builds confidence in their own interpretive skills.

Our work has also further refined what “speaking data” entails for emerging bilingual learners. Based on our work over the past two years, we have identified an array of language learning objectives that undergird learners’ ability to “speak data” (Table 3). The transition from using everyday language to developing facility with the kinds of language used to work with data, highlighted in Table 3, is not intuitive and is challenging to learn, even if practiced in context. We argue this classroom-based data work is a kind of “labor that remains hidden because we are not trained to think of it as labor at all” [36] (p. 192). That the labor is hidden reflects inequities in interpretive and communicative labor; linguistically minoritized learners are expected to work, comprehend, and make use of hard-to-access health information, rather than become the producers of that information [39]. In order to learn how to work with data, learners need opportunities to work with data tools and processes, but, paradoxically, the learner is likely to be evaluated based on the way she participates in those processes, regardless of whether she can participate effectively in English, can use a computer, or work with common digital tools such as Excel. By working with adult learners to name the data inequities in their own lives, and making their interpretive labor more visible, we believe that we are taking foundational steps towards this commitment to communicative justice in data science and public health.

## 5. Conclusions

The women’s learning and discovery processes raise an essential question: How do any of us learn to recognize what to attend to, what to “make meaningful” in these multilingual and multimodal ensembles, much more so during a global health crisis? Integrating data visualization should test popular assumptions about the knowledge-building capacity of groups narrowly defined as “limited English proficient”. The visual message is not just a proxy for written public health messages. Indeed, in the era of digital health, being able to interpret visual information should be viewed as central to digital and health literacy competencies. With greater recognition of the data work underway in community-based adult education classrooms in our conceptual frameworks, research designs, and funding priorities, we will move towards a more complete understanding of the role of language choice (along with other choices we make about modalities and tools) in data visualization/interpretive work.

## Figures and Tables

**Figure 1 ijerph-20-00696-f001:**
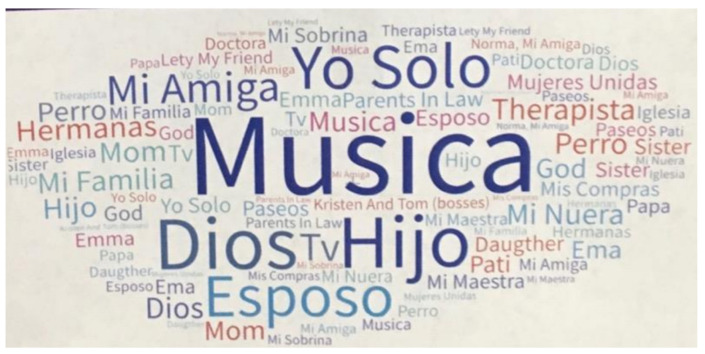
Word cloud of responses to how women found stress relief and social support. Women generated individual responses, and the teachers then converted them into a word cloud. Word clouds, a useful tool for representing data distributions and relative proportions, were a popular form of visualization for this beginner-level English classroom.

**Figure 2 ijerph-20-00696-f002:**
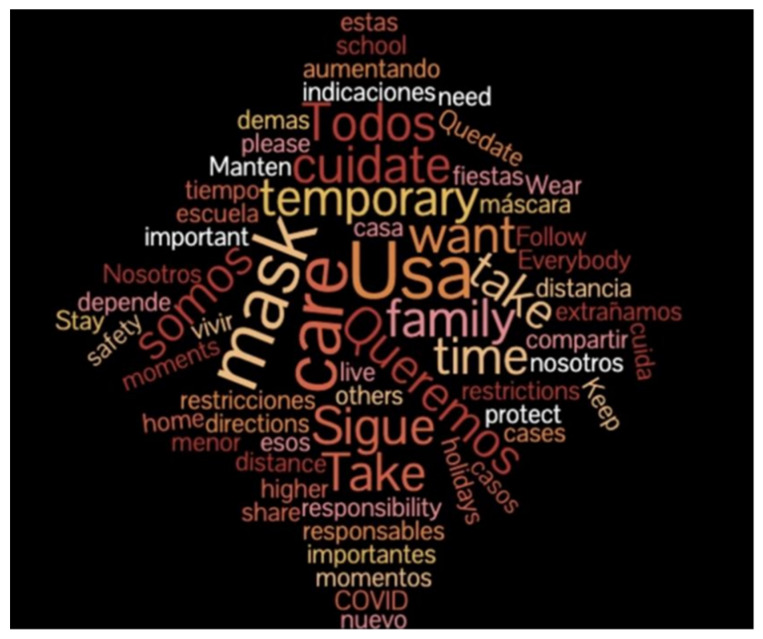
Word cloud summary of the bilingual script of a video created by a learner for her peers to promote mask wearing to prevent COVID-19.

**Figure 3 ijerph-20-00696-f003:**
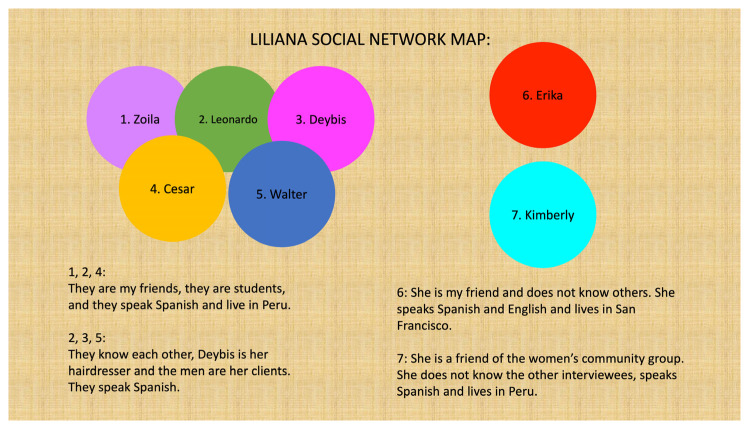
Depiction of a learner’s social network where the learner is not in the image, and people are grouped by their relationships with each other, with two (6 and 7) outside of the main cluster. Names have been changed to preserve confidentiality.

**Figure 4 ijerph-20-00696-f004:**
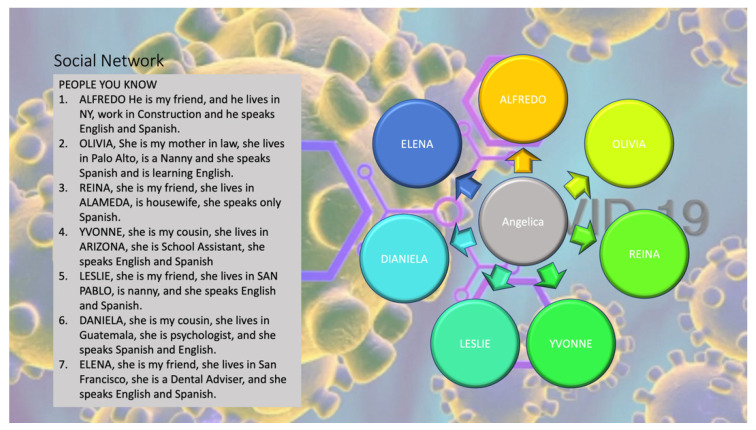
Depiction of a learner’s social network as emanating from herself at the center, even as each name reveals they do not live nearby. Names have been changed to preserve confidentiality.

**Figure 5 ijerph-20-00696-f005:**
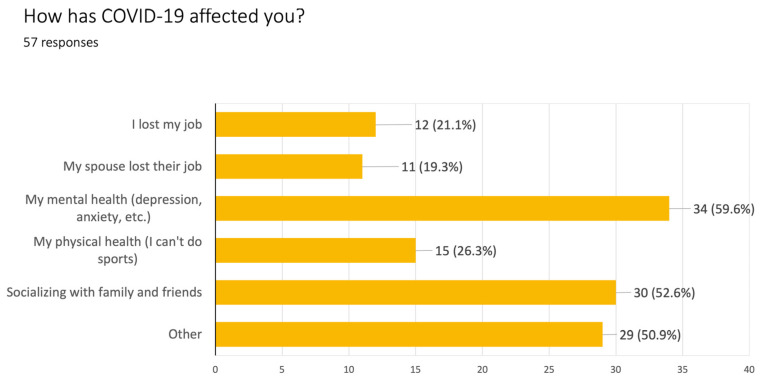
Bar graph (translated from Spanish) based on classroom survey about how COVID-19 is affecting people in their social network.

**Figure 6 ijerph-20-00696-f006:**
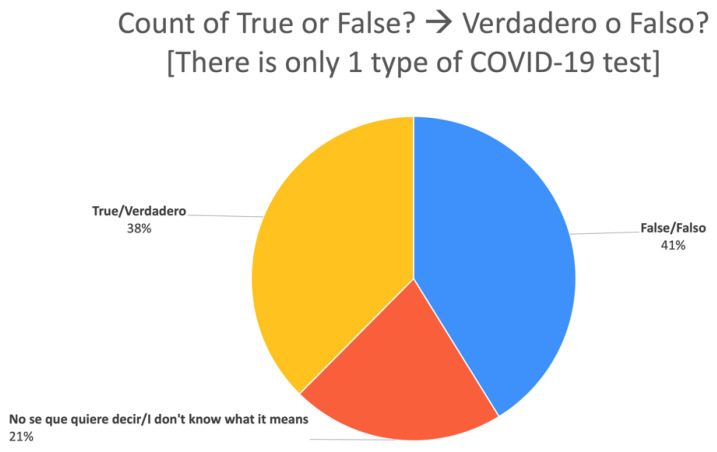
Pie chart representing data from classroom survey of social networks about understanding information about COVID-19 tests.

**Table 1 ijerph-20-00696-t001:** Overview of data visualization work with immigrant Latina women in community-based English classrooms: Sample inquiry questions, data sources, and data visualization tools used to support learner inquiry.

Inquiry Questions Posed by Learners	Types of Data Sources Introduced	Data Visualization Tools Introduced
*Theme: COVID-19 Prevention* What are barriers to social distancing at home and in the workplace during COVID-19? How do I make my home and workplace safer?	*Structural Determinants Data:* Population density, housing costs, job types, unemployment rates, job loss	Social network maps (Figure 3 and Figure 4) Bar graphs (Figure 5)
*Theme: Health and our Neighborhoods* How does where we live affect our health? Why are there so many corner stores in our neighborhood?	*Health data:* Prevalence/mortality/hospitalization data by zip code; prevalence of risk factors (e.g., smoking rates, obesity) *Structural determinants data*: presence/absence of health supportive/limiting policies (liquor store hours; sugar sweetened beverage restriction policies)	Prevalence heat maps
*Theme: Stress and Resilience* What are sources of stress in our community? How do we overcome them?	*Risk Assessment Data:* Risk probability statistics (e.g., how to interpret ‘1 in 7 people’ expressions) *Social Network Data:* Peer and community networks (e.g., diversity of contacts, frequency of contact, language preferences, transnational links)	Word clouds (Figure 1 and Figure 2) Social network maps (Figure 3 and Figure 4) Pie charts (Figure 6)

**Table 2 ijerph-20-00696-t002:** Teaching field notes and examples from the data literacy skill-building activities related to COVID-19 themes, annotated to highlight relevant data interpretation and data visualization skills at work in the classroom.

Fieldnotes for a Series of Classes on Data Literacy Skill-Building	Data Interpretation/ Data Visualizations Skills	Example from Classroom Transcript Analysis	Image from Student Work

Initial Classes focus on ‘what is data? And “What kind of information is useful to me?” Learners were concerned about mask use in their communities. As a class we discussed and tried to find an equivalent in Spanish for “to rank”, we practiced ranking different things and concepts using criteria such as very likely, likely, neutral, unlikely, and very unlikely. Learners also learned to recognize this language when used in Likert scales.	Use language for describing data; Scaled data	Word cloud used to analyze and discuss the role of masks and barriers to wearing masks.	Figure 2

After this initial discussion of Likert scales, the learners and teacher created data questions and data collection tools used Google Forms to design small-scale surveys for gathering information about health knowledge and behaviors in their local communities (e.g., mask-wearing habits, physical activity habits, job-seeking as an essential worker during COVID-19, what people do to advocate for safer working conditions during COVID-19). The learners shared their Google form links with their friends and families.	Create an original data collection tool Identify a need for data	BAR GRAPHS were used to analyze data collected to better understand how COVID-19 had affected the Latino population in the community (Interviewing People about COVID-19 survey, answered by 57 community members in June 2020).	Figure 5

In subsequent lessons, the teacher scaffolded the collaborative interpretation of the data collected and creation of data visuals (e.g., pie charts, bar graph, and lists). The learners practiced asking and answering questions for data interpretation, such as questions for description (e.g., “What percentage of people answered ‘likely’? How many people is that?”), questions for prediction (e.g., “How do you think people will answer in this question?) Now, let’s look at their answers. (Are their answers different or similar to your predictions?”), and questions for interpretation (e.g., “Why do you think people answered this way? Do these answers reflect what my own community thinks?”). These dialogues gave learners opportunities to practice generating responses such as: “This answer was interesting because 75% of people said they were very unlikely to go out without a mask outside, but there was a 25% that said they would. It’s interesting because masks are required.”	Identify different kinds of data visuals Ask good questions Use language for describing and interpreting data	Analysis of data visualizations in both English and Spanish (transcribed from a Zoom lesson, August 2020).	Figure 5 and Figure 6

**Table 3 ijerph-20-00696-t003:** Language learning objectives that support “speaking data” in data science (adapted from D’Ignazio & Bhargava, 2016).

Practice asking good questions (yes/no, open questions) that help learners clarify understanding of a data set and act on their curiosity
2. Describe inquiry and discovery processes, using phrases associated with sorting, categorizing, naming, doubting, etc.
3. Describe data sources and data visuals (e.g., visual, outlier, categories, pie graph, line graph, phrases such as, “this line shows” or, “this color represents”.)
4. Describe differences in shape, center, and spread in data visuals, including outlier data, using adjectives and adverbs, and related expressions for comparison/contrast
5. Interpret data and data visuals, using phrases for comparing/contrasting, expressing change over time, and making predictions
6. Show respect for different interpretations about data, using expressions associated with collaborative dialogue
7. Express preference about data interpretations and visuals using expressions for expressing learning styles and preferences
8. Express belief, doubt, hope, shock, anger, grief, and other emotions in response to data using a range of emotion vocabulary
9. Describe possible biases and assumptions, including one’s own, when talking about data, using expressions and vocabulary associated with critical inquiry and inclusive language

## Data Availability

Not available.

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
