# Peer review of "Working with Data in Adult English Classrooms: Lessons Learned about Communicative Justice during the COVID-19 Pandemic"

_ijerph, 2022, doi:10.3390/ijerph20010696_

Round 1
Reviewer 1 Report
Insufficient theoretical framework, with inadequate aspects (name an author and own experience in this section). There is not enough information about the participants. very simple analysis Very simple results and representation.
Author Response
- Reviewer comment: Insufficient theoretical framework, with inadequate aspects (name an author and own experience in this section).
Response
- We have added text to clarify the main content of the theoretical approach, which is Social Constructivist Theory, and added text with references for it.
- Under 2.1 Pedagogical Approach, we refer to the theoretical model of Social Constructivist Theory (“From a social constructivist perspective, data literacy is best viewed as an interactional competence [25,26]; data literacy tools and processes become accessible to adult learners through their re-occurring interactions with those tools and processes with others.”.
- We added text as well in the Methods to further indicate the theory underlying the work (lines 138-142)
- The classes taught by María José Bastías (third author) incorporate a participatory approach to data literacy and data visualization that emphasizes shared inquiry into data sources and creative interpretive elements (e.g., using role play to explore different interpretations of data) and reflect a social constructivist approach to how knowledge is created, , which we have used in our classroom-bases work previously”
- We did not understand what this phrase “with inadequate aspects (name an author and own experience in this section)" means as a recommendation to change the paper, but have inferred it relates to the need for references and our use of this theory in the past.
- We have added text to clarify the main content of the theoretical approach, which is Social Constructivist Theory, and added text with references for it.
- There is not enough information about the participants.
Response
- We do not agree that more description of the participants is needed. What is provided is as follows:
"The classroom demographics reflected the community affiliated with the non-profit Latinx organization; the women were originally from Central America, predominantly from Mexico, El Salvador, and Guatemala. Most worked in service industries as child, patient, and senior caregivers, janitors, restaurant workers, housekeepers, homemakers, among others. The majority of the women (75%) were between 20-49 years old, with another 25% between 50-70 years old." - We are characterizing an ESL setting for which age group, gender, language, employment and country of origin are presented. Our ESL partners have preferred we do not collect and report individual-level data on participants.
- We do not agree that more description of the participants is needed. What is provided is as follows:
- Very simple analysis
Response- We have added some additional analysis of the language-based interactions that accompany the data visualization process and have added a sentence to the methods. Line 193-195 (see following response)
"For several of the classes, we analyzed recorded transcripts to create word clouds (Figure 1 & 2) and to provide examples of the translanguaging collaborative work by students."
- We have added some additional analysis of the language-based interactions that accompany the data visualization process and have added a sentence to the methods. Line 193-195 (see following response)
- Very simple results and presentation
Response
- Figures are presented as they were prepared in the classroom. We have added additional data in the form of figures of results from student work, which are referred to in Table 2 and added as figures 5 and 6 with discussion of these results added in the text.
-
Figures 5 and 6 have been added, with the following text in Results: line 310
“Figures 1, 2, 5, and 6 depict examples of data visuals that integrate learner-generated data. In Figure 5, a survey administered by students to members of their social network about How Covid-19 has affected them, led to a bar graph of the main results and discussion of the concepts as well as the results. For example, the bar graph represents all responses and answers are not mutually exclusive so total greater than 100%. Students then explored how to take responses from the bar graph and transform it into a word cloud to compare the different modes of presentation (data not shown)."
-
- Figures are presented as they were prepared in the classroom. We have added additional data in the form of figures of results from student work, which are referred to in Table 2 and added as figures 5 and 6 with discussion of these results added in the text.

Reviewer 2 Report
The reviewed study reports on the methodology used for language learning courses offered for adults, in particular women in the lantinx community, and the lessons learned from their application, specifically in the context of the shifting modality to online classes and other difficulties caused by the covid-19 pandemic.
The description of the work is thorough, but I fell that it lacks some kind of evaluation that allows the reader to get a better grasp of how the learning objectives set out for the students are actually achieved. Something more than the recollection of the experience made by the teachers. This shortcoming is indeed mentioned by the authors as a future endeavour, but I think, at least some basic measures, should be included in this study.
I also fail to recognize much novelty in the lessons learned metioned in section 4. This is my personal interpretation of them:
1. Paragraph 1 relates to the importance of collaborative learning and the importance of context when aquiring new knowledge. Significant learning is more easily achieved when provided in context to the learners' previously acquired knowledge.
2. Entry barriers need to be accounted for before significant learning can be achieved (in this case technological barriers)
3. I have a special concer about this lesson, as per mentioned in line 423: "We have found that the women's most incisive interpretations of covid-19 data visuals are not fixed to any one language; rather it appears that the movement across laguanges and modalities strengthens their risk comprehension...". I am not sure I understand how this can be deduced from the experiences reported, as I would have expected to see multi-language evidence of the interpretations, whilst collected examples (other than the word clouds) tend to be all presented in spanish.
4. Learning by doing is a core strategy for significant learning. In this sense, I completely understand the necessity of developing the capacity to 'speak data'.
Perhaps a better way (in my opinion) to convey these messages would be to clearly state the lesson and how it should be applied in adult second language education at the beginning of each paragraph, to then proceed with the basis for the finding.
There are some issues with the Figures. Figure 2 appears before fig. 1, and also the caption on figures seem to be cropped. For example, fig. 2's caption ends like 'changed to preserve'; to preserve what?. Captions also appear to be cropped for figures 1 and 3.
Author Response
- Reviewer comment: [...] it lacks some kind of evaluation that allows the reader to get a better grasp of how the learning objectives set out for the students are actually achieved. Something more than the recollection of the experience made by the teachers. This shortcoming is indeed mentioned by the authors as a future endeavour, but I think, at least some basic measures, should be included in this study.
Response- We recognize the importance of evaluation and have made an effort to include student outcomes in the form of work completed that represents the concepts applied since we were not able to collect evaluation data during the Zoom classes for privacy reasons. We think the strengths in our results are adequate, we clarify the content created, the lesson concepts, and the engagement in both created visuals and in interpretive text from classroom participants
-
Reviewer comment: I also fail to recognize much novelty in the lessons learned mentioned in section 4.
Response:- We have to respectfully disagree. We think there are many emerging results from this work. We added this sentence to the beginning of Section 4 to more directly specify two of the elements we think add to the literature.
“To our knowledge, this is the first study exploring the unique constellation of making meaning from a range of emerging and often conflicting data during a pandemic that also applies translanguaging and collective participatory data analysis.”
- We have to respectfully disagree. We think there are many emerging results from this work. We added this sentence to the beginning of Section 4 to more directly specify two of the elements we think add to the literature.
- Reviewer comment: I have a special concer about this lesson, as per mentioned in line 423: "We have found that the women's most incisive interpretations of covid-19 data visuals are not fixed to any one language; rather it appears that the movement across laguanges and modalities strengthens their risk comprehension...". I am not sure I understand how this can be deduced from the experiences reported, as I would have expected to see multi-language evidence of the interpretations, whilst collected examples (other than the word clouds) tend to be all presented in spanish.
Response
- Learners collaboratively drafted questions about covid in their communities in both English and Spanish. Then they surveyed their community on the topics. The class then had a discussion focused on analyzing: topic of the question (risk prevention, perception, behavior), language choice (selecting most suitable words to avoid misunderstanding from interviewees), considering modality and language (proficiency in first and second language as well as interviewee’s familiarity with use of technology), and predicting possible answers.
- We also ADDED a bilingual text excerpt (lines 325-381) that displays the interrogation of information in English and Spanish about numbers, words to depict relative concepts like ‘more’ and ‘most’ as follows which accompanies Figure 6 in the paper:
Figure 6 represents a pie chart from the same student-led survey and conveys the results about views on what types of COVID-19 tests to trust, which led to a discussion about the relationship between words that convey size (many, majority, some, few) and different ways these might be seen in numeric data as percentages or frequencies and what level of detail is enough to make a decision about taking a specific type of test. The following exchange represents the way the discussion of data interpretation is providing an opportunity for multilingual and multimodal interpretation of information.
- Reviewer comment: Learning by doing is a core strategy for significant learning. In this sense, I completely understand the necessity of developing the capacity to 'speak data'. Perhaps a better way (in my opinion) to convey these messages would be to clearly state the lesson and how it should be applied in adult second language education at the beginning of each paragraph, to then proceed with the basis for the finding.
Response- We have provided more specific guidance, while at the same time acknowledging that each classroom may need to use a flexible approach this in a different way (as we describe in our pedagogical approach). Table 2 now has more details in the first column descriptions, which can then be applied to other topics. For example, row 1 starts now with, “Initial Classes focus on ‘what is data? And “What kind of information is useful to me?” to provide guidance on how to start.
- Row 2 then begins with “After this initial discussion of Likert scales, the learners and teacher created data questions and data collection tools used Google Forms to design small-scale surveys for gathering information about health knowledge and behaviors in their local communities” and
- Row 3, “In subsequent lessons, the teacher scaffolded the collaborative interpretation of the data collected and creation of data visuals (e.g., pie charts, bar graph, and lists).”
- Reviewer comment: Figure formatting issues.
Response- Corrected problems with figure order, numbering, and captions.

Reviewer 3 Report
1. This study focuses on the use of data visualizations as health risk communication tools to solve specific populations commonly described as “limited English proficient”. And it proposed a series of data literacy pedagogy. Overall, this study has a considerable contribution and reference value.
2. Row number 201, develop counter-narratives to the widespread COVID-19 misinformation they were encountering in their daily lives. It is recommended to enumerate what misinformation about COVID-19 is included.
3. It is advisable to briefly introduce the theoretical background of degree centrality and isolate in social networks, and further explain what teachers teach learners about social networks and how to apply them.
4. It is recommended to draw a research flow chart so that readers can quickly understand the process of the entire research implementation, and to strengthen the elaboration of research methods, including Observation and Action research.
5. Figure 1 and Figure 2 are placed in the wrong order. And Figure 3 does not seem to be displayed perfectly, it is recommended to update to a clearer picture.
6. Table 1 shows the data visualization tools introduced to support learner inquiry. Where are Bar graphs, Prevalence heat maps, and Pie charts used?
Author Response
- Reviewer comment: Row number 201, develop counter-narratives to the widespread COVID-19 misinformation they were encountering in their daily lives. It is recommended to enumerate what misinformation about COVID-19 is included.
Response- We added text to clarify the misinformation examples (lines 209-214) that were the focus of at least one of the classroom sessions. We also added Figure 6 which refers to the trust about different COVID tests
"Beginning in March 2020 and extending for the next 9 months, María José shifted to teaching online modules lasting 3-5 weeks focused on learners’ desire to understand COVID-19 trends and develop counter-narratives to the widespread COVID-19 misinformation they were encountering in their daily lives, such as that there were no requirements of employers to provide masks or to use appropriate ventilation in close workspaces."
- We added text to clarify the misinformation examples (lines 209-214) that were the focus of at least one of the classroom sessions. We also added Figure 6 which refers to the trust about different COVID tests
- Reviewer comment: It is advisable to briefly introduce the theoretical background of degree centrality and isolate in social networks, and further explain what teachers teach learners about social networks and how to apply them.
Response- We have added text to clarify the main content of the theoretical approach, which is Social Constructivist Theory, and added text with references for it.
- Under 2.1 Pedagogical Approach, we refer to the theoretical model of Social Constructivist Theory (“From a social constructivist perspective, data literacy is best viewed as an interactional competence [25,26]; data literacy tools and processes become accessible to adult learners through their re-occurring interactions with those tools and processes with others.”.
- We added text as well in the Methods to further indicate the theory underlying the work (lines 138-142)
- The classes taught by María José Bastías (third author) incorporate a participatory approach to data literacy and data visualization that emphasizes shared inquiry into data sources and creative interpretive elements (e.g., using role play to explore different interpretations of data) and reflect a social constructivist approach to how knowledge is created, , which we have used in our classroom-bases work previously”
- We have added text to clarify the main content of the theoretical approach, which is Social Constructivist Theory, and added text with references for it.
- Reviewer comment: It is recommended to draw a research flow chart so that readers can quickly understand the process of the entire research implementation, and to strengthen the elaboration of research methods, including Observation and Action research.
Response
- We added headers to the Results section to address this need for clarification to distinguish more between Results in the Student Led work and those Results that related to the assessment of the data.
- 3.2 Student-Led Social Network Surveys and Data Visualizations (line 307)
- 3.3 Assessment of Classroom Examples (line 424)
- We added headers to the Results section to address this need for clarification to distinguish more between Results in the Student Led work and those Results that related to the assessment of the data.
- Reviewer comment: Figure formatting issues
Response- Corrected formatting, ordering, and numbering of figures
- Reviewer comment: Table 1 shows the data visualization tools introduced to support learner inquiry. Where are Bar graphs, Prevalence heat maps, and Pie charts used?
Response- We have added additional figures (5 & 6), for bar graphs and pie charts. We also have several more, and can include these but were not sure if there were limits to the number of figures so we kept it at 6.

Round 2
Reviewer 1 Report
The authors have made the requested changes